# Experience of “One Stop TB Diagnostic Solution” Model in Engaging a Private Laboratory for End-to-End Diagnostic Services in the National TB Elimination Program in Hisar, India

**DOI:** 10.3390/diagnostics13172823

**Published:** 2023-08-31

**Authors:** Rajesh Raju, Banuru Muralidhara Prasad, Umesh Alavadi, Sanjeev Saini, Mukesh Sabharwal, Akshay Duhan, Sridhar Anand, Manohar Lal, Harpreet Kaur, Neerja Arora, Jyoti Jaju, Moe Moore, Ranjani Ramachandran, Nishant Kumar, Rajendra P. Joshi

**Affiliations:** 1Directorate of Health Services, Government of Haryana, Swasthya Bhavan, Sector 6, Panchkula 134109, Indiadtohrhsr@rntcp.org (M.S.); dr.akshayduhan@yahoo.in (A.D.); 2Infectious Disease Detection and Surveillance (IDDS), New Delhi 110058, India; 3United States Agency for International Development, New Delhi 110021, India; umeshalavadi@yahoo.com; 4TB Support Network, Office of the World Health Organization (WHO) Representative to India, WHO Country Office, New Delhi 110021, India; 5iDEFEAT Project, IQVIA, New Delhi 110021, India; 6iDEFEAT Project, The International Union against Tuberculosis and Lung Disease (The Union), New Delhi 110016, India; 7Office of the World Health Organization (WHO) Representative to India, WHO Country Office, New Delhi 110021, India; 8Central TB Division, Ministry of Health and Family Welfare Government of India, New Delhi 110001, India

**Keywords:** tuberculosis, diagnostic algorithm, drug-resistant tuberculosis, private laboratory, turnaround time, one-stop model

## Abstract

The complete diagnostic evaluation of tuberculosis based on its drug-resistance profile is critical for appropriate treatment decisions. The TB diagnostic landscape in India has been transformed with the scaling-up of WHO-recommended diagnostics, but challenges remain with specimen transportation, completing diagnostic assessment, turnaround time (TAT), and maintaining laboratories. Private laboratories have demonstrated efficiencies for specimen collection, transportation, and the timely testing and issue of results. A one-stop TB diagnostic model was designed to assess the feasibility of providing end-to-end diagnostic services in the Hisar district of Haryana state, India. A NTEP-certified private laboratory was engaged to provide the services, complementing the existing public sector diagnostic services. A total of 10,164 specimens were collected between May 2022 and January 2023 and these were followed for the complete diagnostic assessment of Drug-Susceptible TB (DS-TB) and Drug-Resistant TB (DR-TB) and the time taken for issuing results. A total of 2152 (21%) patients were detected with TB, 1996 (93%) Rifampicin-Sensitive and 134 (6%) with Rifampicin-Resistant TB. Nearly 99% of the patients completed the evaluation of DS-TB and DR-TB within the recommended TAT. The One-Stop TB/DR-TB Diagnostic Solution model has demonstrated that diagnostic efficiencies could be enhanced through the strategic purchase of private laboratory services.

## 1. Background

Tuberculosis has received global attention and a political commitment to end TB by 2030 through the United Nations high-level meeting of the General Assembly on the fight against tuberculosis [1]. India is one of the high TB burden countries and has committed to ending TB by 2025 through strategies envisaged in the “National Strategic Plan” with a focus on building and strengthening the health system for early detection and appropriate treatment [2]. Early detection through high-sensitivity diagnostics and universal access to quality TB diagnosis is one of the key priority areas. The diagnostic network under the National TB Elimination Program (NTEP) has expanded, with molecular diagnostics at districts and sub-district level (5090 sites with Xpert/Truenat) and TB culture and drug-susceptibility testing facilities (83 Culture and Drug Susceptibility Testing (DST), 34 intermediate reference laboratories) with cutting-edge technologies at state and national levels (six national reference laboratories) [3]. The diagnosis of Drug-Susceptible TB (DS-TB) and Drug-Resistant TB (DR-TB) requires sequential testing and these tests are available at different levels of the health system. At times, completing the diagnostic assessment of identified presumptive TB patients (PTBPs) could be a challenge and may result in the loss of patients along the care cascade [4]. An incomplete diagnostic assessment could also lead to inappropriate treatment resulting in the amplification of drug resistance [5].

The NTEPs diagnostic network and the strategies to increase access to diagnostic services have faced challenges, some of which are: (a) specimen collection and transportation from facilities to different laboratories; (b) the time taken to complete the diagnostic assessment tests required to rule out or detect drug resistance, which are carried out at different facilities at different locations; and (c) the provision of quality-assured diagnostic coverage for patients in the private sector [2]. Though the program has strengthened the district and reference laboratories, running laboratories is resource intensive and requires the procurement of laboratory reagents, consumables, and test kits, maintenance, and calibration of equipment, sustaining human resources and their competencies, and implementing quality assurance measures [6]. An option to engage private sector laboratories to complement and enhance the diagnostic services of the public sector is explored as part of the strategy [7].

Private sector laboratories have demonstrated efficiencies through a person-centric approach for specimen collection, the transportation, testing, and reporting of results in the desired turnaround time (TAT) for a range of hematological and biochemical tests, operationalized through a hub-and-spoke model. Many of the private laboratories have been accredited by the National Accreditation Board for Testing and Calibration Laboratories (NABL) [8]. The NTEP has also certified 14 laboratories for providing Line Probe Assay (LPA) and Culture DST diagnostic tests [3]. The efficiency of private laboratories is harnessed using the “turn-key model” of the National AIDS Control Program (NACP), where specimens for viral load testing were collected from district sites and tested at a centralized laboratory and reports are shared in the desired TAT. Recently, the government has strategically engaged several private-sector laboratories for enhancing diagnostic services during the COVID-19 pandemic [9].

Drawing from these experiences of private laboratory engagement in public health programs, and with the understanding that the TB diagnostic assessment involves multiple tests and the results of the preceding test determine the need for subsequent testing, the “One-Stop TB/DR-TB diagnostic solution” model was designed. This innovative model was conceptualized with the objectives to assess the feasibility, impact, and cost implications of the strategic purchase of TB testing services from a private laboratory for end-to-end operations—specimen collection, transportation, performing tests as per NTEP’s algorithm—and sharing results within the desired TAT. The article here outlines the model’s concept, implementation, and learnings, which could be transitioned into other high-burden TB countries.

### 1.1. The One-Stop TB Diagnostic Solution Model

The One-Stop TB Diagnostic Solution Model was implemented by the Infectious Disease Detection and Surveillance (IDDS) Project supported by the United States Agency for International Development (USAID) under the guidance of Central TB Division, Ministry of Health and Family Welfare, Government of India. In the model design, the healthcare providers (public/private) identify and refer the PTBPs/or diagnosed TB patients for assessment. The specimens are collected and transported from the point of collection to a centralized NTEP-certified private laboratory where requisite tests are performed. Along with the upfront Nucleic Acid Amplification Test (NAAT), the model offers first- and second-line Line Probe Assay (FL-LPA and SL-LPA) and Liquid Culture and Drug Susceptibility Testing (LC-DST), as per the NTEP’s diagnosed algorithm [3]. The Chest X-ray (CXR) services for PTBPs are also offered through facilities that are engaged locally by the contracted private laboratory. Pre-treatment evaluation tests for diagnosed DR-TB patients are offered as part of the overall service delivery package. The PTBPs are registered at respective health facilities in the NTEP’s web portal (Ni-kshay) and test reports are updated as and when available. The results are also communicated to the program staff and the referring physician via email, for treatment decisions.

### 1.2. Model Implementation

The model was implemented in the Hisar district of Haryana in collaboration with the District Health Administration and District TB Office (Figure 1). The district has an approximate population of two million with a mix of urban, semi-urban, and rural populations and an estimated TB prevalence of 465 (326–605) persons per 100,000 population [10]. The district has one District TB Centre (DTC), eight TB Units (TUs), 20 designated microscopy centers, and two NAAT (GeneXpert and Truenat) facilities at the DTC and at the medical college. The average distance of the designated microscopy centers to the district headquarters is 28 km (min 7 km–max 59 km). The district is linked to the Intermediate Reference Laboratory (IRL) at Karnal (182 km) for FL-LPA, SL-LPA, and LC-DST, and to National Reference Laboratory at Delhi (118 km from Karnal) for LC-DST of certain second-line drugs.

An NTEP-certified private laboratory (Thyrocare Technologies Private Ltd., Mumbai, India) was selected through an open procurement process and was contracted for providing diagnostic services as envisaged in the model. In consultation with the district NTEP staff, route plans were prepared for specimen collection from public health facilities (tuberculosis units, designated microscopy centers, health and wellness centers, subcenters, and primary healthcare facilities) and private facilities (clinics and hospitals) and the CXR facilities engaged were also mapped (see Appendix A). The contracted laboratory deployed runners who covered nearly 80 facilities (36 public and 44 private) across the district. The runners reach each facility as scheduled daily to collect the specimens. Upon reaching the facility, staff hand over the collected specimens (two specimens for each patient) to runners along with test request forms (TRF) which have a Ni-kshay ID generated for the patient. The runners deposit the specimens at the district hub from where it is transported to the Xpert testing facility at Gurugram the same day (which is about 200 km from Hisar). Following an Xpert result of mycobacteria tuberculosis (MTB) being detected, the second specimen is flown the next day to the NTEP-certified laboratory based in Mumbai for FL-LPA, SL-LPA, and Liquid Culture Drug Susceptibility Test (LC DST) (see Appendix A). The tests are performed in compliance with the NTEP diagnostic algorithms [3]. The test results are entered as and when they are available in the Ni-kshay portal against the IDs available in the TRF. The referring physicians at respective facilities in Hisar could view the results on the portal and initiate the treatment appropriately. In addition, the laboratory also shared reports via email with the health facilities at the district and sub-district levels.

## 2. Methods

### 2.1. Data Source

This is a cohort study where the patient’s specimen data for the period from 16 May 2022 to 31 January 2023 were collected through the TRF provided by the respective facilities along with specimens. These data were entered in an Excel-based project (Microsoft Office 2019 version) recording and reporting sheet to ensure all specimens complete the diagnostic care cascade. The data were verified with the Ni-kshay portal by district NTEP staff and the implementing IDDS team. The reporting variables, like date of specimen collection, type of specimen collected, type of patients (PTBP or diagnosed TB patients), diagnostic tests performed, and issuance of results, were also collected.

### 2.2. Analysis and Statistics

The data analysis is carried out using the SPSS 16.0 software (IBM SPSS Statistics, Armonk, NY, USA). All specimens with valid results generated through the model are considered output variables. For the outcome analysis:(a)Specimens with Xpert negative results are considered MTB negative and no further tests are conducted;(b)Specimens with Xpert positive results (MTB detected and rifampicin-sensitive) and FL-LPA (rifampicin and isoniazid-sensitive results) are considered DS-TB;(c)Specimens with the following Xpert results: MTB detected and rifampicin-resistant results; with FL-LPA (rifampicin and/or isoniazid-resistant results); SL-LPA (showing resistance to fluoroquinolones); and LC-DST (results showing resistance to any of the following drugs: moxifloxacin, linezolid, and pyrazinamide) are considered DR-TB. The analysis is presented in the result section as per the tests performed;(d)All specimens are tracked from the point of collection to the issuance of results, as mentioned above, for an analysis of turnaround time (TAT).

## 3. Results

### 3.1. Demographic and Clinical Profile of Patients

A total of 10,164 patient specimens were collected and processed during the model implementation period, out of which 9468 (93%) specimens were collected from PTBPs for the diagnosis of TB and the remaining were from diagnosed TB patients for further assessment for drug resistance (Table 1). More than 88% of the specimens collected were from public facilities and 11% were from private facilities. The private facilities provided specimens for >50% of the extrapulmonary specimens. The maximum number of specimens were collected from rural areas rather than urban areas and from male patients and from patients ≤ 60 years.

### 3.2. Completing TB Diagnostic Algorithm

The specimens collected were processed using GeneXpert (Xpert MTB/rifampicin) for the diagnosis of TB and 21% of specimens resulted in MTB being detected (Table 2). Excluding the specimens from diagnosed TB patients for further testing for drug resistance, MTB was detected in 1457 (15%) specimens from PTBPs (Table 2). Rifampicin resistance (RR) was detected in 5.7% of PTBP MTB positive specimens (84 patients) and 7% of specimens from diagnosed TB patients. Almost 100% of Xpert MTB specimens were subjected to FL-LPA. There were 13 specimens with insufficient quantity for processing that were requested for repeat samples for further testing and were not received. Out of the total of 2136 specimens, 1838 were smear-positive and 298 were smear-negative prior to performing FL-LPA tests (Figure 2). All smear-negative specimens were inoculated for Liquid Culture and 93 showed growth (31%). A total of 227 specimen results showed resistance to one of the first-line drugs (rifampicin/isoniazid).

Following the NTEP diagnostic algorithm, 227 specimens were processed for SL-LPA. The results showed that 37 specimens were resistant to fluoroquinolones and 4 specimens were found to be resistant to fluoroquinolones (FQ) and second-line injectable drugs (SLID). The remaining 186 specimens were sensitive to both FQ and SLID. All 227 specimens were processed for LC in a Mycobacteria Growth Indicator Tube (MGIT 960); out of these 133 showed growth and the DST results for 128 specimens were available. The results for the remaining specimens are expected shortly, at the time of manuscript preparation. A total of 11 specimens showed resistance to moxifloxacin, 2 specimens were resistant to linezolid, and 41 were resistant to pyrazinamide. Resistance to moxifloxacin and pyrazinamide is seen in five specimens and resistance to linezolid and pyrazinamide was seen in two specimens.

### 3.3. Patients Completing Drug-Susceptible TB Diagnostic Algorithm

The specimens with Xpert results of being MTB positive and rifampicin-sensitive (RS) were eligible to be tested for FL-LPA. Prior to FL-LPA tests, smear tests were performed. The results of smear tests determined whether the specimen was to be processed directly for LPA or indirectly following LC (MGIT 960). Out of the 1996 eligible for FL-LPA, 1715 (86%) had smear-positive results and 267 (13%) were smear-negative. The smear-negative specimens were subjected to LC and 12 specimens were rejected for insufficient specimen quantity and 2 repeat specimens were not received. Including the culture-positive samples (89/267, 33%), 1806 specimens were tested for FL-LPA (including two smear-negative and culture-negative samples processed for LPA). The tests were not conducted for 174 specimens with culture-negative results and 14 specimens for which repeat specimens were not received (Table 3). Among the specimens tested for FL-LPA, 93% were sensitive to both rifampicin and isoniazid and were declared as DS-TB. The specimens resistant to isoniazid, rifampicin, and both isoniazid and rifampicin were 108 (6%), 1 (<1%), and 5 (4%), respectively, and these 114 specimens were subjected for SL-LPA and LC-DST as per the algorithm.

Out of the 114 specimens eligible for SL-LPA, 99 (87%) were sensitive to FQ and SLID. About 13% of the specimens were resistant to either FQ or SLID. The culture DST results were available for 108 specimens and results were anticipated for 6 specimens while the manuscript was being prepared. The LC-DST results showed resistance to moxifloxacin (3/108, 3%) linezolid (1/108, 1%), and pyrazinamide (14/108, 13%). Two specimens were resistant to both moxifloxacin and pyrazinamide and one specimen was resistant to linezolid and pyrazinamide. None of the specimens were resistant to both fluoroquinolone and linezolid. Almost 99% of the cohort of patient specimens with MTB positive/RS Xpert test results completed the diagnostic algorithm.

### 3.4. Patients Completing Drug-Resistant TB Diagnostic Algorithm

Out of the total 134 RR TB specimens, 109 were smear-positive and processed for FL-LPA directly, and the remaining 23 were subjected to LC (2 specimens were excluded, 1 repeat specimen and 1 for insufficient quantity). Two specimens with culture-positive results were processed for FL-LPA. Among the total 111 specimens processed for FL-LPA, isoniazid and rifampicin resistance was seen in 81 (73%), and rifampicin resistance was seen in 26 (23%) specimens (Table 4).

A total of 107 specimens resistant to either isoniazid or rifampicin following the FL-LPA results were tested for SL-LPA. Among the specimens tested, 81 (76%) were sensitive to FQ and SLIDs and 26 (24%) specimens were resistant to either of the drugs. In the model, all those specimens (107 specimens) with any drug-resistant results were subjected to LC-DST. The LC results for three specimens were still under process. The results were available for 104 specimens, which showed 8% resistance to moxifloxacin (8/104), 1% to linezolid (1/104), and 25% to pyrazinamide (26/104). The resistance to both moxifloxacin and pyrazinamide was seen in three specimens and one specimen showed resistance to both linezolid and pyrazinamide. Overall, 96% of patient specimens completed the DR-TB diagnostic algorithm in the model.

### 3.5. Turnaround Time for Completing Diagnostic Tests

The TAT analysis included the time interval between the day of specimen collection to the issuance of results. The TAT is shown for respective tests along the diagnostic care cascade (Table 5). The median number of days for reporting results for Xpert, FL-LPA, SL-LPA, and LC-DST was 1 day, 5 days, 7 days, and 45 days, respectively. For specimens with smear-negative results requiring liquid culture prior to LPA, the results were shared within 30 days.

The TAT for RS and/or DS-TB specimens with smear-positive results was about 4–6 days, including results for FL-LPA. For RR and/or DR-TB patients, the TAT was in the range of 6–9 days, including SL-LPA. However, with the inclusion of LC for smear-negative and LC-DST for certain drugs of DR-TB patients, the median TAT for completing the NTEP diagnostic algorithm up to pyrazinamide susceptibility testing was 45 days. For all liquid cultures, negative results were declared within 42 days, as per the protocol for declaring results in MGIT 960.

## 4. Discussion

The salient features of the innovative “One-Stop TB diagnostic solution model” are: (a) the staff at the health facility generated a Ni-kshay ID for all PTBPs and two specimens were collected for testing; (b) the runners reached facilities as scheduled along the designated route maps covering rural and urban areas to collect the specimen and deposit it at the district hub, from where the logistic team shipped them to the Xpert lab and NTEP-certified culture-DST lab of the contracted private laboratory; (c) the NTEP diagnostic algorithm was followed and the results were updated as and when they were available in the portal, which could be accessed by referring to the facility; (d) a one-time specimen is collected and it is ensured that all tests are performed as per the algorithm; (e) the results are issued as per the NTEP-prescribed TAT for respective tests; and (f) the laboratory services in the public sector continued to perform Xpert and other tests, complementing the diagnostic services for the district.

The findings from the model could be referred to two studies from India. A cohort study conducted by Uma Shankar et al. in Karnataka involved a review of program records and the mapping of sequential testing [12]. In Uma Shankar’s study, patients with Xpert MTB-detected results were traced to a reference laboratory and the results showed that FL-LPA reports were available for 71% of patients. However, SL-LPA reports were available for 20% of eligible and culture results for 40% of patients. Overall, 95% of patients with Xpert RS results completed the algorithm. In the same study, for those with RR specimens, FL-LPA reports were available for 55%, SL-LPA for 31%, and 49% completed the diagnostic algorithm. Comparing these results with the Hisar model, 99% of specimens with RS and 98% RR completed the diagnostic algorithm (a comparative table with the model is available in Appendix A). The “DOST” intervention model in Delhi was designed to strengthen the patient care pathway, collected 9331 specimens of patients from private providers, and transported them to Xpert sites in public facilities through an interface agency [13]. The Xpert results showed 382 specimens were RR and 198 (52%) specimens were subjected to SL- LPA, out of which 134 (68%) completed the tests. The LPA tests were also performed in the IRL of the NTEP. Both studies were conducted in routine programmatic settings with the involvement of public health laboratories. However, in the one-stop diagnostic solution model, the services were provided by a contracted private laboratory.

A complete TB diagnostic assessment is critical for a clinician to decide on an appropriate treatment regimen. Empirical treatment regimens, due to the non-availability of certain test results, may cause an amplification of drug resistance, treatment failures, and the continued transmission of infection [14]. Along with completing the diagnostic assessment, it is equally important to obtain results within the stipulated TAT. The model has demonstrated that one-time specimens collected from patients, and issuing results within the stipulated TAT, would reduce the need for multiple visits to the same/different providers or visits to higher centers. The availability of timely test results and prompt initiation of appropriate treatment could also reduce transmission. The indicated TAT for various tests includes collection, performing the tests, and issuing the results of the program, and is 1–2 days for Xpert and 2–4 days for LPA [15]. In the model, these TATs were achieved, which were much lower compared to other studies from Delhi—2.6 days for Xpert, 26 days for LPA—and for the study from Karnataka—1 day for Xpert, 5 days for LPA (see Appendix A for a comparison of the TAT of NTEP with the studies and model) [12].

The pre-treatment evaluation (PTE) of diagnosed DR-TB patients was part of the overall diagnostic service package included in the model. The diagnosed DR-TB patients were counseled and assessed by the nodal officer who prescribed the evaluation tests. All the tests were reported within a day to the medical officer for the initiation of the appropriate treatment regimen. In the model, sequential testing was also completed for twenty-two specimens with Xpert Rifampicin Indeterminant results (see Appendix A).

The model has demonstrated the fact that diagnostic and operational efficiencies could be incorporated into routine programmatic settings to strengthen diagnostic services. The runner concept could facilitate ensuring specimens are picked and reach the facility as scheduled and in a timely manner, and the services could be bundled along with other laboratory services for sustainability. The laboratory operations could be extended beyond regular timings with a 6/8 hourly shift basis for laboratory technicians to provide results on the same day and to ship specimens to higher centers for further testing. The program may also explore the purchase of services from private laboratories, as demonstrated in the model.

Overall, the model is designed to complete the diagnostic care cascade and no efforts were made for additional case finding. This is one of the limitations of the model and others worth mentioning are: (a) the CXR services included in the model had a negligible role in the diagnostic care cascade; and (b), though the runners were available for transport, repeat specimens were not available for 14 patients, and the team coordinated with the program and other partners to trace the patients. However, these were lost to follow-up due to multiple reasons. Limited efforts were made by the private laboratory to address the discordant results for ten specimens between FL-LPA and Xpert by performing a third test on Xpert for RR status. However, this discordance was resolved for four specimens.

## 5. Conclusions

The TB diagnostic services are complex and require sequential testing to complete the diagnostic care cascade. The services are available at multiple locations along the diagnostic network, which could cause delays in completing tests and the subsequent initiation of appropriate treatment. The one-stop TB/DR-TB diagnostic solution model is the only model to date that has demonstrated the end-to-end operations to the completion of the NTEP diagnostic care cascade for TB and the TAT which could be achieved by engaging a private laboratory. The learnings from the model suggests exploring the strategic purchase of services from NTEP-certified private laboratories.

## Figures and Tables

**Figure 1 diagnostics-13-02823-f001:**
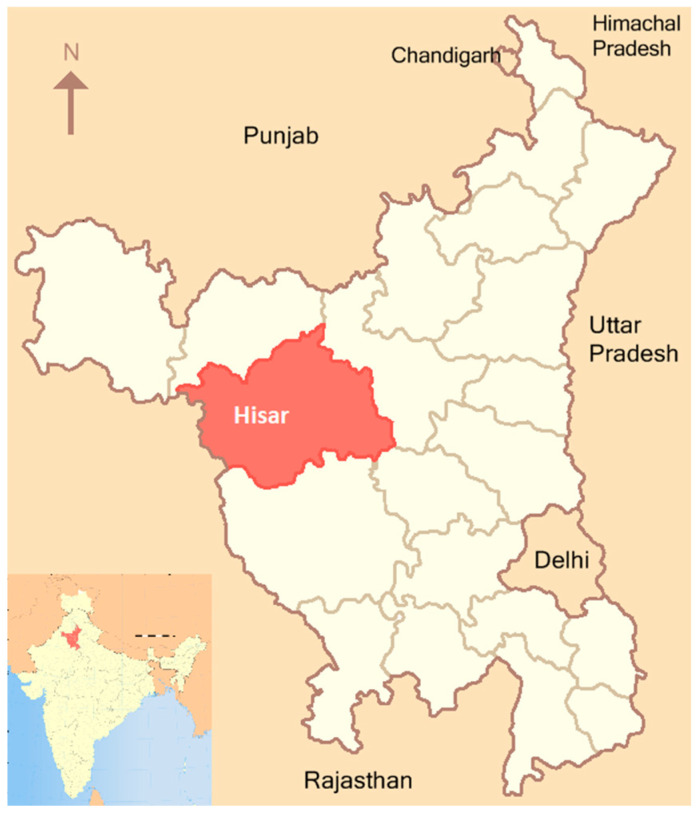
Map of India showing Hisar District in Haryana State. Reprinted from reference [11].

**Figure 2 diagnostics-13-02823-f002:**
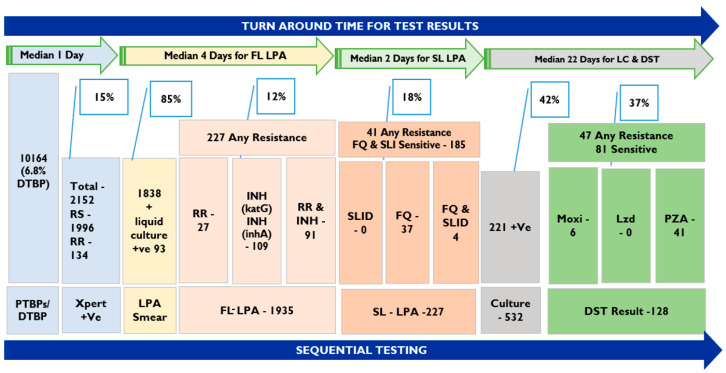
Specimens completing TB diagnostic tests as per NTEP algorithm and median turnaround time under the One-Stop TB Diagnostic solution model. Abbreviations: PTBPs—presumptive TB patients, DTBP—Diagnosed TB Patients, RS—rifampicin sensitive, RR—rifampicin resistance, INH—isoniazid, LPA—Line Probe Assay, FL-LPA- first line LPA, SL-LPA—second line LPA, SLID—second-line injectable drugs, FQ—fluoroquinolones, +Ve—positive, Moxi—moxifloxacin, Lzd- Linezolid, PZA—pyrazinamide. LC—liquid culture, LC and DST—liquid culture and drug susceptibility testing, DST—drug susceptibility tests.

**Table 1 diagnostics-13-02823-t001:** Demographic and clinical profile of patients’ specimens collected in the model.

Variable	Number	Percentage
Total Specimens in the Model	10,164	
Type of patient		
PTBPs	9468	93.2%
Diagnosed TB Patients	696	6.8%
Type of Referring Facility		
Public	9005	88.6%
Private	1159	11.4%
Type of Specimen		
Pulmonary	9924	97.6%
Extra Pulmonary	240	2.4%
Geography		
Rural	6660	65.5%
Urban	3504	34.5%
Gender		
Male	6663	65.5%
Female	3500	34.4%
Transgender	1	
Age (years)		
≤ 14	284	2.8%
15–29	2249	22.1%
30–44	2309	22.7%
45–59	2411	23.7%
≥ 60	2911	28.6%

**Table 2 diagnostics-13-02823-t002:** Results of upfront specimens tested using Xpert machines in the model.

Xpert Result for MTB	Number	Percentages
MTB Detected	2152	21.2%
MTB Not Detected	8012	78.8%
Rifampicin Status Xpert		
Rif-Resistant	134	6.2%
Rif-Sensitive	1996	92.7%
Rif Indeterminate	22	1.0%

Rif—Rifampicin.

**Table 3 diagnostics-13-02823-t003:** Patients completing Rifampicin-Sensitive TB Diagnostic Algorithm.

	Rifampicin-Sensitive (RS) TB Diagnostic Algorithm	One-Stop TB Model	
sl no.	Particulars	Number	Percentage
1	Number of RS TB Patients diagnosed using Xpert	1996	
2	Number of samples that reached the reference lab and were eligible	1996	100%
3	FL-LPA carried out with results	1806	90%
a	H- and R-Sensitive	1692	94%
b	H-Resistant and R-Sensitive	108	6%
c	R-Resistant only	1	<1%
d	H- and R-Resistant	5	<1%
4	FL-LPA not done	2	
5	SL-LPA done	114	100%
6	SL-LPA not done	0	0
i	FQ- and SLI-Sensitive	99	87%
ii	FQ- and/or SLI-Resistant	15	13%
7	Liquid Culture—DST done	114	100%
8	Result anticipated for LC-DST (Moxi, PZA, Lzd)	6	5%
9	Result for LC-DST (Moxi, PZA, Lzd)	Moxi (3/108), Lzd (1/108), PZA (14/108).	
10	Number of Rifampicin-Sensitive patients completing the diagnostic algorithm	1985	99%

Moxi—moxifloxacin, Lzd—linezolid, PZA—pyrazinamide, H—isoniazid, R—rifampicin, FL-LPA—first line LPA, SL-LPA—second line LPA, FQ—fluroquinolone. RS—rifampicin sensitive. SLI—second line injectable. LC-DST—liquid culture drug susceptibility test.

**Table 4 diagnostics-13-02823-t004:** Patients completing Rifampicin-Resistant TB Diagnostic Algorithm.

	Rifampicin-Resistant TB Diagnostic Algorithm	One-Stop TB Model	
sl no.	Particulars	Number	Percentage
1	Number of RR-TB patients diagnosed on Xpert testing	134	
2	Number whose samples reached the C-DST lab for further cascade testing	134	
i	FL-LPA carried out	111	100%
ii	H- and R-Sensitive	4	3%
iii	R-Resistant	26	23%
iv	H- and R-Resistant	81	73%
3	FL-LPA not carried out	2	1%
4	Liquid Culture carried out	23	17%
	Liquid Culture positive (included into FL-LPA tests)	2	8%
5	SL-LPA carried out	107	96%
i	FQ- and SLI-sensitive	81	76%
ii	FQ- and/or SLI-Resistant	26	24%
6	SL-LPA Not carried out	2	2%
7	Liquid Culture DST carried out	107	100%
i	Results anticipated for Liquid Culture DST	3	5%
ii	Result for LC-DST (Moxi, PZA, Lzd)	Moxi (8/104) Lzd (1/104) PZA (26/104).	
8	Number of Rifampicin-Resistant patients completing the diagnostic algorithm	132	98%

Moxi—moxifloxacin, Lzd—linezolid, PZA—pyrazinamide, H—isoniazid, R—rifampicin, FL-LPA—first line LPA, SL-LPA—second line LPA, FQ—fluroquinolone. RR—rifampicin resistance. SLI—second line injectable. LC-DST—liquid culture drug susceptibility test.

**Table 5 diagnostics-13-02823-t005:** Turnaround time (TAT) for completing tests in model.

Test Name *	Pre-Lab TAT	Lab TAT	Total TAT
Xpert test	1	1	1 (1–1)
FL-LPA	n.a	4 (3–5)	4 (3–5)
SL-LPA	n.a	2 (1–4)	2 (1–4)
LC-DST **	n.a	22 (17–32)	22 (17–32)

n.a—not applicable, *—median and interquartile range (IQR), FL-LPA—First Line LPA, SL-LPA—Second-Line LPA, LC-DST—Liquid Culture Drug susceptibility test. **—following LC Test positive result.

## Data Availability

The data pertaining to the model are analyzed and presented in the manuscript. The data are part of the routine program and reported in Ni-kshay portal. This may be available upon request to concerned officials of the National TB Elimination Program.

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
