# Peer review of "Experience of “One Stop TB Diagnostic Solution” Model in Engaging a Private Laboratory for End-to-End Diagnostic Services in the National TB Elimination Program in Hisar, India"

_diagnostics, 2023, doi:10.3390/diagnostics13172823_

Round 1

Reviewer 1 Report

An important topic, a public health problem that remains current (and will certainly remain after 2030), an article that brings information to those interested.

However, there are things to improve for a publishable article. I don't know if it's the fatigue accumulated over the year and the lack of a real vacation, but the material is very difficult to read. Not all readers are interested in the problem of tuberculosis, but even an interested reader might be a little confused and give up reading to the end.

The idea of ​​putting more items in additional tabs is welcome, however the article remains difficult to read. It needs to be examined by native English speakers to clarify the topic of the phrases but it also needs a careful revision to eliminate any typographical errors.

Please apply the rule that each abbreviation entered appears linked to the first occurrence of the abbreviated word in the text (both in the abstract and in the entire manuscript).

Once an abbreviation has been stated, it is not necessary after only a few lines to write down the word(s) in full again and bracket the abbreviation again.

Also pay attention to the use of the hyphen and the position of a space before/after it.

The introduction could be shortened.

In subpoint 1.2, model implementation it is true that there is a "supplement F1/F2" association, but a clear graphic showing the work-flow should be drafted and attached to the main text. It should not be an "overloaded" figure but one that allows a quick assessment by the reader.

Table 1 requires either changing the title or, perhaps better, separating it into different tables.

In fact, all the tables are very busy. A table, graph should represent a visual aid for the reader and not an element to spend many minutes on to understand its meaning and usefulness. All tables should be re-evaluated and redone in this regard.

Both in the discussions and especially in the conclusions, the reader must be helped to understand very clearly the meaning and usefulness of the proposed model and the involvement of private laboratories. In the current form of the manuscript this is not obvious.

In the conclusions subchapter, it is not clear which conclusions derive from the study carried out by the authors and which from already known data from other studies. The conclusions appear too general and do not prove to the reader that the reality is the one discussed. The conclusions subsection needs to be redone and improved.
The manuscript could be prepared to become publishable.

The topics (in the current form) does not help the reader to easy understand how important the subject is.
Please request the support from an English native speaker/writer for improvement.

Author Response

Authors Response:

We thank the reviewer for going through the article. As you have rightly put forward, the article is too technical and only those who are in Tuberculosis diagnostic area could possibly understand better than the general audience. We authors have reviewed and have avoided technical jargons to make it understandable for healthcare providers in India and globally.

Your suggestion to redo the table 1 is well accepted and we have split the table 1 into table 1 & 2. We have also reworked on the abbreviations in the entire manuscript and the hyphen in the manuscript.

Yes, we understand that the introduction section is lengthy. The introduction section is setting the context to TB diagnostic services in India. Many would have limited understanding of the services and how the network is functioning. Secondly, there was also a need to highlight the efforts made by NTEP towards expanding molecular diagnostic services and strengthening the diagnostic network.

We understand that the model implementation infographic picture is currently placed as a supplement. As you rightly pointed that they are heavily loaded and therefore we have placed in supplement. The infographic picture clearly explains the process and authors were of the agreement to keep the picture in the current format.

We attempted to convert the table into flow diagram and we believed that the percentages could be lost and may limit the message that we would like to convey to the readers. The flow diagram is placed below in the document for your reference.

The discussion and conclusions are updated as per the suggestions by the reviewer. However, too much emphasis on the model could also be a limiting factor as key stakeholders are part of the NTEP.

Figure 1 Patients completing Rifampicin Sensitive TB Diagnostic Algorithm

Figure 2 Patients completing Rifampicin Resistant TB Diagnostic Algorithm.

We appreciate the reviewers comments and we hope that that all the comments are addressed.

with regards

Prasad

===========================================

Reviewer 2 Report

Title:

It’s too long

Background/ Introduction:

·        Acknowledge the work of other researchers on the topic, (Diagnosis of Drug susceptible TB (DS TB) and Drug resistant TB ( DR TB), identify the gaps in knowledge which need to be filled

·        Describe clearly General and specific objectives of your study

·        Describe selection criteria followed to engage private lab (Thyrocare Technologies Private Ltd , Mumbai)

·        WHO were runners responsible for transportation of specimen?, Weather they were trained ?

·        How they were paid (to ensure cost effectiveness)

Methods: 

·        Describe the study site and the study population

·        Describe the data collection instrument ( test request form TRF)

·        Who filled the forms ? Whether the filled forms were checked for completeness

Results:

Only 11% of specimen were collected from private health facilities, although more patients seek care from private health facilities in comparison to public health facilities 

Discussion:

·        Discussion section is weak, requires improvement

·        There is no need to describe to describe “ one stop TB diagnostic model in discussion section

·        Describe  important findings of your study

·        Summarize key findings and give your interpretation

·        Discuss implications

·        Acknowledge limitations of your study

·        Make appropriate recommendations

Conclusion:

·        Restate your study problem

·        Summarize main points and describe the significance of results

no comments

Author Response

We thank the reviewer for going through our manuscript and the comments are much appreciated. Hope we have addressed all the comments satisfactorily.

Background/ Introduction:

  1. Acknowledge the work of other researchers on the topic, (Diagnosis of Drug susceptible TB (DS TB) and Drug resistant TB ( DR TB), identify the gaps in knowledge which need to be filled

Authors response:

Many thanks for going through the lengthy article. We understand that setting the context for someone who is not aware of TB diagnostic network and also highlight the efforts of NTEP was also required. There are limited article that follow the TB diagnostic pathway, like the way demonstrated in the model.

  1. Describe clearly General and specific objectives of your study

Authors response:

The general objectives of the model are now presented in the lines 81 to 85 and the specific objective is mentioned in line 85 and 86.

  1. Describe selection criteria followed to engage private lab (Thyrocare Technologies Private Ltd , Mumbai)

Authors response:

Please see the lines 117-119 in the manuscript which explains that the laboratory was selected through an open procurement process.

  1. WHO were runners responsible for transportation of specimen?, Weather they were trained ? How they were paid (to ensure cost effectivenss).

Authors response:

The runners were basically employed by the private laboratories. They were deployed to collect the specimens from the facilities along the route map prepared in consultation with district TB officer, NTEP staff and notifying private doctors. They were paid on daily basis for completing each route irrespective of number of specimens collected.

Methods: 

  1. Describe the study site and the study population

Authors response:

We may please request you to kindly see the lines from 106 to 115. Here we explain the study site and study population.

  1. Describe the data collection instrument ( test request form TRF). Who filled the forms ? Whether the filled forms were checked for completeness

Authors response:

The model was implemented under the guidance and integration with NTEP. The data was collected from test request forms (TRFs) duly filled by the laboratory technician or any other person as designated by the facility in-charge. The test request forms have programme Ni-kshay ID (government portal) which contains the specific details of patients. The programme staff reviewed the data in the portal and updated the missing information (if any). The private laboratory updated the result for each of the specimens collected against the Ni-kshay ID (this is a unique number given for each registered patients).

Results:

  1. Only 11% of specimen were collected from private health facilities, although more patients seek care from private health facilities in comparison to public health facilities.

Authors response:

The services of the model were provided as an option to the existing diagnostic services in the district for providers, either public or private. Who so ever referred the specimens of PTBPs/diagnosed TB patients were included in the model and end-to-end sequential tests were performed as per NTEP diagnostic algorithm.

Discussion:

  1. Discussion section is weak, requires improvement

Authors response:

We understand that the available literature in the diagnostic services areas is limited and there is a challenge to substantiate the findings from the model.

  1. There is no need to describe to describe “ one stop TB diagnostic model in discussion section

Authors response:

We have removed the describing the model again and have summarized the model points.

  1. Describe  important findings of your study; Summarize key findings and give your interpretation

Authors response:

We may please request you to kindly refer to lines 285 to 294 where we have listed the outputs from the model.

  1. Discuss implications

Authors response:

We may please request you to kindly refer to lines 315 to 328 where we have discussed about the implications of delayed diagnosis and treatment initiation.

  1. Acknowledge limitations of your study

Authors response:

We may please request you to kindly refer to lines 344 to 351 where we have listed the limitations and how we attempted to addressed the same.

  1. Make appropriate recommendations

Authors response:

We may please request you to kindly refer to lines 335 to 342 where we have listed the appropriate recommendations.

Conclusion:

  1. Restate your study problem
  2. Summarize main points and describe the significance of results

Authors response:

The conclusion section is updated as per the reviewer’s suggestion.

===========================================

Reviewer 3 Report

Dear Editor

This article reported one stop TB diagnostic solution model by engaging a private laboratory for

end-to-end diagnostic services in the national TB elimination program, which is an interesting

and important topic today. However, there are some minor concerns in this article that need to be

addressed.

1.  The abbreviation of each word should be followed by the full name, (e.g. RIF, etc.). Please kindly correct it in the text.

2.  Replace “Sensitive and Resistant” to “sensitive and resistant”.

3. Please write the names of the antibiotic in small letters.

4. How is the sample size determined? Please provide the calculation formula.

5. Please include the summary of culturing and testing for antibiotic resistance in the methods section.

6.  The inclusion and exclusion criteria in the methods section has not been defined..

7. Add the meaning of MTB, TB, and PTBPs under the Figure 1 for better understanding.

8. The discussion section should be improved to show the importance of the findings.

9. The strengths and limitation of the study should be mentioned..

10. References 14, 15, and 16 were not referred in the manuscript. Please insert them.

Dear Editor

This article reported one stop TB diagnostic solution model by engaging a private laboratory for

end-to-end diagnostic services in the national TB elimination program, which is an interesting

and important topic today. However, there are some minor concerns in this article that need to be

addressed.

1. For the first time, the abbreviation for the word should be followed by the full name, and

then just enter the abbreviation (e.g. RIF, etc.). Please kindly correct it in the text.

2. Please change “Sensitive and Resistant” to “sensitive and resistant”.

3. Please write the names of the antibiotic in small letters.

4. How is the sample size determined? Please provide the calculation formula.

5. Please include the summary of culturing and testing for antibiotic resistance in the

methods section.

6. Please include the inclusion and exclusion criteria in the methods section.

7. Please add the meaning of MTB, TB, and PTBPs under the Figure 1 for better

understanding.

8. Please improve the discussion section.

9. Please state the strengths of this study.

10. References 14, 15, and 16 were not referred in the manuscript. Please insert them.

Author Response

We thank the reviewer for going through the manuscript and appreciate the comments provided to improvise. We hope that we have addressed all the comments satisfactorily.

  1. The abbreviation of each word should be followed by the full name, (e.g. RIF, etc.). Please kindly correct it in the text.

Authors response: We thank you for the observation, we have updated the same.

  1. Replace “Sensitive and Resistant” to “sensitive and resistant”.

Authors response: We thank you for the observation, we have updated the same.

  1. Please write the names of the antibiotic in small letters.

Authors response: We thank you for the observation, we have updated the same.

  1. How is the sample size determined? Please provide the calculation formula.

Authors response: The one-stop TB diagnostic solution model was implemented in the Hisar district of Haryana. All PTBPs identified in the district either in the public facility or private facility could deposit the specimens at respective collection centers for availing the diagnostic services. This is a population based intervention model and the sample size was not determined.

  1. Please include the summary of culturing and testing for antibiotic resistance in the methods section.

Authors response: The one-stop TB diagnostic solution model only collected the specimens and transported to NTEP certified private laboratory. As part of the NTEP certification, the process of culturing and testing for antibiotic resistance is being followed and reviewed by National Reference Laboratory of the TB programme regularly.

  1. The inclusion and exclusion criteria in the methods section has not been defined..

Authors response: The one-stop TB diagnostic solution model was implemented in the Hisar district of Haryana and the specimens available from any PTBP was considered for analysis.

  1. Add the meaning of MTB, TB, and PTBPs under the Figure 1 for better understanding.

Authors response: We may please request you to kindly refer to the lines 207 to 212 for the expansion of abbreviations.

  1. The discussion section should be improved to show the importance of the findings.

Authors response: We may please request you to kindly refer to the lines 336 to 343 in the discussion section.

  1. The strengths and limitation of the study should be mentioned..

Authors response: We may please request you to kindly refer to the lines 285-295 for strengths of the model and 344 to 352 lines for the limitations.

  1. References 14, 15, and 16 were not referred in the manuscript. Please insert them.

Authors response: Many thanks for the observation and there were two references which was cited previous removed in the track change mode. However was in the library and reflected in the refence list. The same has been updated.